# Nonreplicating Adenoviral Vectors: Improving Tropism and Delivery of Cancer Gene Therapy

**DOI:** 10.3390/cancers13081863

**Published:** 2021-04-14

**Authors:** Nayara Gusmão Tessarollo, Ana Carolina M. Domingues, Fernanda Antunes, Jean Carlos dos Santos da Luz, Otavio Augusto Rodrigues, Otto Luiz Dutra Cerqueira, Bryan E. Strauss

**Affiliations:** Viral Vector Laboratory, Center for Translational Investigation in Oncology, Cancer Institute of São Paulo/LIM24, University of São Paulo School of Medicine, São Paulo 01246-000, Brazil; nayara.tessarollo@hc.fm.usp.br (N.G.T.); ana.domingues@fm.usp.br (A.C.M.D.); fernanda.antunes@hc.fm.usp.br (F.A.); jc.santosluz@usp.br (J.C.d.S.d.L.); otavio.rodrigues@hc.fm.usp.br (O.A.R.); ottoluix@usp.br (O.L.D.C.)

**Keywords:** nonreplicating adenovirus vector, cancer, gene therapy, routes of delivery, virus coated with cancer cell membrane

## Abstract

**Simple Summary:**

The treatment of cancer has progressed greatly with the advent of immunotherapy and gene therapy, including the use of nonreplicating adenoviral vectors to deliver genes with antitumor activity for cancer gene therapy. Even so, the successful application of these vectors may benefit from modifications in their design, including their molecular structure, so that specificity for the target cell is increased and off-target effects are minimized. With such improvements, we may find new opportunities for systemic administration of adenoviral vectors as well as the delivery of strategic antigen targets of an antitumor immune response. We propose that the improvement of nonreplicating adenoviral vectors will allow them to continue to hold a key position in cancer gene therapy and immunotherapy.

**Abstract:**

Recent preclinical and clinical studies have used viral vectors in gene therapy research, especially nonreplicating adenovirus encoding strategic therapeutic genes for cancer treatment. Adenoviruses were the first DNA viruses to go into therapeutic development, mainly due to well-known biological features: stability in vivo, ease of manufacture, and efficient gene delivery to dividing and nondividing cells. However, there are some limitations for gene therapy using adenoviral vectors, such as nonspecific transduction of normal cells and liver sequestration and neutralization by antibodies, especially when administered systemically. On the other hand, adenoviral vectors are amenable to strategies for the modification of their biological structures, including genetic manipulation of viral proteins, pseudotyping, and conjugation with polymers or biological membranes. Such modifications provide greater specificity to the target cell and better safety in systemic administration; thus, a reduction of antiviral host responses would favor the use of adenoviral vectors in cancer immunotherapy. In this review, we describe the structural and molecular features of nonreplicating adenoviral vectors, the current limitations to their use, and strategies to modify adenoviral tropism, highlighting the approaches that may allow for the systemic administration of gene therapy.

## 1. Overview: Structural and Molecular Features of Nonreplicating Adenoviral Vectors

Adenoviruses (Ads) are one of the most well-studied and widely used viral vectors, representing 17.5% (*n* = 573) of vectors used in gene therapy clinical trials [1]. The first gene therapy was approved in 2003 by the China Food and Drug Administration. Gendicine is a recombinant nonreplicating adenovirus encoding human p53 and, despite more than 17 years of commercial use, it has only been tested in clinical trials in China for the treatment of hepatocellular, nasopharyngeal, gastric, liver, lung, breast, prostate, ovarian, and head and neck cancer, either alone or in combination with radio- or chemotherapy [2]. This therapy serves to illustrate the potential for using nonreplicating adenoviral vectors as part of effective cancer treatment.

Recently, nonreplicating adenoviral vectors have gained attention due to their use in the development of vaccines, especially to combat SARS-CoV-2, the novel coronavirus. These vaccines include those based on recombinant adenovirus serotypes, such as human adenovirus vector 5 [3,4,5,6,7], chimpanzee adenovirus vector ChAdOx1 [8,9], and combined human serotypes vectors 5 and 26 [10,11]. The Ad vectors can elicit robust and durable cellular and humoral immune responses [12]. The induction of a balanced innate immune response makes the Ad vectors good candidates for vaccine platforms, and they can also play a role in cancer gene therapy.

Replicating adenoviral vectors are used for the induction of oncolysis (also referred to as oncolytic adenovirus or virotherapy), and these vectors have played a major role in showing the potential of adenoviruses in cancer immunotherapy. The use of nonreplicating adenoviral vectors also deserves particular attention. We argue that nonreplicating vectors perform quite well and may even offer advantages when compared to the use of their replicating counterparts, especially concerning the delivery of proimmune but antiviral transgenes. While we do not discount oncolytic viruses, we do support the continued development of nonreplicating adenoviral vectors for cancer immunotherapy. As shown in Table 1, many clinical trials that are underway involve the use of nonreplicating adenoviral vectors for cancer gene therapy. In this review, we focus on nonreplicating adenoviral vectors, discussing vector biology and current barriers to cancer gene therapy. We also propose some strategies that enhance vector performance, especially in terms of virus delivery and targeting, thus supporting the use of nonreplicating adenoviruses for cancer immunotherapy.

Human adenoviruses (HAds) are subdivided into seven species (A–G) and >50 serotypes based on serological properties, DNA homology, genome organization, and oncogenicity [24]. More than 100 types of human adenovirus and >200 nonhuman Ad serotypes have been identified to date [25].

Ads carry a linear double-stranded DNA genome (26 to 46 kb in length) and core proteins inside an icosahedral capsid [26]. The Ad DNA genome contains two inverted terminal repeats (100–140 bp) and can be divided into five early genes and five late genes. The capsid facets are formed by structural proteins, mainly composed of hexons, and each vertex contains a penton base that anchors the trimeric protein fiber, divided into the fiber knob and shaft. The viral particles have around one million amino acid residues (weight around 150 MDa) and an average size of 90–100 nm [27]. In the infection process, the knob interacts with cell surface receptors such as the coxsackievirus and adenovirus receptor (CAR), CD46, CD80/86, and desmoglein 2 (DSG2) [28,29]. This interaction leads to viral particle immobilization, which facilitates the interaction between the penton base and integrins [30] and, thus, virion internalization. The DNA and some core proteins are transported through the microtubular complexes to the nuclear pore and are introduced into the cell’s nucleus [31]. Inside the nucleus, the viral DNA remains episomal, and the expression of the early genes (E1A, E1B, E2, E3, and E4) suppresses transcription from the host genome, thus favoring adenovirus protein synthesis and replication. Then, the late genes (L1–L5) are expressed, leading to virus encapsulation and viral particle maturation in the nucleus during the completion of the lytic cycle. Nuclear and cytoplasmatic membranes are disrupted, and new virions are released from permissive cells 48–72 h after infection [32,33].

Many characteristics of adenoviruses, such as their safety, broad cell tropism, and ability to stimulate a robust immune response, favor their use as a viral vector platform employed as a gene delivery tool in gene therapy, as an oncolytic cancer treatment, and in the development of vaccines [12,32]. Moreover, the genome of Ads is well characterized, genetically stable, and does not integrate into the host’s genome but remains as episomal DNA in the cell nucleus. In addition, adenoviral vectors are modified to control viral replication, have a large cloning capacity (up to 37 kb), can transduce both dividing and quiescent cells, and have high in vivo transduction efficiency [34,35,36]. Human adenovirus type 5 (HAd5) is the most frequently used adenovirus for the development of gene therapy vectors, which promote the expression of transgenes in the target cells yet have impaired replication and, hence, prevent unwanted virus spread. There are three generations of nonreplicating adenoviral vectors used in gene therapy. In the first generation, E1 and E3 early genes were deleted, rendering vector replication defective but maintaining the ability to transduce host cells without killing them and liberating ~8 kb of space in the genome for the genetic payload (transgene(s) plus regulatory sequences) [37,38]. Since the E1 region is essential for virus replication, E1A proteins induce the transcription of Ad genes, and E1B proteins inhibit cellular apoptosis. Vector production requires that the E1 gene be supplied by transcomplementation, using cell lines (such as HEK293 or PERC.6) that were modified to incorporate the viral E1 region [39,40]. For the second-generation adenoviral vectors, beyond E1/E3 deletions, E2 or E4 regions have also been removed, providing additional space for cargo sequences (~10.5 kb). Third-generation adenoviral vectors were generated after deletion of almost all viral sequences except for the ITRs, the packaging signal, and minimal sequences required for genome replication and encapsulation during vector production [41].

Therefore, nonreplicating adenoviral vectors, different from their replicating counterparts, do not provoke the same cellular responses due to their lack of viral protein expression, absence of viral genome replication, and deficiency in the ability to induce cytopathic effects.

## 2. Current Applications of Nonreplicating Adenoviral Vectors in Cancer Immunotherapy

There are two main routes to delivering gene therapy vectors: ex vivo and in vivo. In vivo gene transfer raises concerns related to the specificity of vector transduction and transgene transcription in the intended target cells in order to achieve the desired therapeutic outcome, a goal that may be compromised by off-target effects. Ex vivo gene transfer occurs outside the body, where the patient’s cells are modified and reinfused. Here, we focus on the in vivo route, particularly the challenges associated with the antiviral immune response.

In general, gene therapy approaches that overcome the immunosuppressive tumor microenvironment (TME) and activate an antitumor immune response are expected to function as cancer immunotherapies. To this end, adenoviral vectors have been modified with a variety of immune-stimulating genes, such as cytokines, costimulatory molecules, tumor-associated antigens, and tumor-suppressor genes [42]. The purpose is not only the direct killing of tumor cells but also the activation of immune cells to attack the tumor. Thus, gene therapy may induce immunogenic cell death and/or the liberation of factors that will then go on to promote the immune response.

The adenoviral vector itself is expected to participate in the activation of an antiviral response that may be both an asset and a complication for gene therapy since attracting the immune response to the tumor site is desirable but the inhibition of viral activity may thwart treatment. After viral entry, pathogen-associated molecular patterns (PAMPs), including viral nucleic acids and viral capsids, are recognized by pattern recognition receptors (PRRs) and activate antiviral immune responses that result in the production of type-I interferons (IFNs), proinflammatory cytokines, and chemokines.

Another important signaling cascade stimulated by the interaction of the virus with CAR and αv integrins is nuclear factor-kB (NF-kB), which mediates the expression of chemokines and interleukin (IL)-1 [43]. Inside the cell, viral DNA is sensed by several cytosolic PRRs such as Toll-like receptor (TLR)-9 [44], DNA-dependent activator of IFN-regulatory factors (DAIs) [45], cytosolic inflammasomes (NALP3) [46], and nucleotide-binding oligomerization domain-like receptors (NOD-like receptors (NLRs)) [47]. As a result, a signaling cascade is initiated, either dependent or independent of myeloid differentiation primary response gene 88 (MyD88), which culminates in the transcription factor (NF-kB, IRF3, IRF7)-mediated expression of IFN-α, IFN-β, and IL-6, among other proinflammatory cytokines and chemokines. In turn, the immunosuppressive TME is modulated to facilitate the recruitment of antigen-presenting cells (APCs) and helper and cytotoxic T-cells. Adenoviral vectors can be especially useful in the treatment of cold tumors [48], which lack immune infiltrate, although the increase in immune cell infiltration may not be enough for the eradication of the tumor [49]. Thus, the approach may be improved if the vector is armed with additional immune-stimulating factors.

Replication-deficient adenoviral vectors have been employed as vaccines and in cancer gene therapy due to strong humoral and T-cell responses to transgenes expressed by the vector [50]. Tatsis et al. (2007) showed that the application of replication-defective adenoviral vectors resulted in sustained levels of CD8^+^ T-cells specific for the transgene product and persistent levels of transcriptionally active adenoviral vector genomes at the site of inoculation in the liver and lymphatic tissues [51]. In comparisson, replicating adenoviral vectors mimic natural virus infection, resulting in the induction of cytokines and costimulatory molecules that provide a potent adjuvant effect [52]. Both nonreplicating and replicating Ad vectors have been shown to activate effector CD8^+^ T-cells and central memory T-cells in treated mice. For this, Osada et al. [53] compared Ad5[E1+]CEA, a replicating adenoviral vector carrying the carcinoembryonic antigen (CEA) with two nonreplicating vectors, Ad5[E1−] and Ad5[E1−, E2b−]. When used for the ex vivo transduction of human dendritic cells (DCs), they found that all three vectors yielded similar infectivity and temporal dynamics of transgene expression. In addition, replicating Ad5[E1+]CEA showed toxicity to DCs, eliciting less maturation of DCs and greater clearance by NK cells. Moreover, Ad5[E1−] and Ad5[E1−, E2b−] were superior to Ad5[E1+] in their capacity to induce and expand antigen-specific T-cell responses. The results suggest that increased replication of an Ad vector may result in diminished efficacy in this scenario, and the deletion of E1, E2, and E3 genes promoted a superior generation of CEA-specific T-cell responses in mice with pre-existing Ad5 immunity.

In the following discussion, we highlight some of the strategies for using nonreplicating adenoviral vectors as cancer immunotherapies in preclinical and clinical assays.

For example, preclinical outcomes in a prostate cancer model have revealed the benefit of immunotherapy based on a heterologous prime-boost, where the virus is injected as a vaccine with concomitant administration of a PD-1-blocking antibody. Similarly, a ChAdOx1–MVA vaccination strategy (a simian adenovirus, ChAdOx1, with the modified vaccinia Ankara virus, MVA) induced CD8^+^ T-cell responses to the tumor-specific self-antigen of prostate 1 (STEAP1) in murine models. The combination with the anti-PD-1 antibody improved the survival of the animals since tumors were abolished in 80% of the mice [54].

Our laboratory has developed an adenoviral vector, AdRGD-PG, with improved tropism and transgene expression. By including the RGD motif in the fiber knob, transduction no longer relies on CAR but instead uses integrins, which are more widely distributed. The use of a p53-responsive promoter (called PG) to control transgene expression resulted in high-level expression in the presence of wild-type p53 [55,56]. When used to deliver p19Arf (a functional partner of p53) and IFN-β, we observed cooperation between these genes for the induction of cell death in vitro and in vivo using the mouse model of melanoma, B16–F10 [56,57]. Moreover, only combined gene transfers conferred the emission of immunogenic cell death markers ATP, calreticulin, and HMGB1 [56]. The combination of p19Arf and IFN-β proved to be an effective immunotherapy since we confirmed the participation of natural killer (NK) cells and CD4^+^ and CD8^+^ T-lymphocytes in immune protection against B16-F10 tumor progression [58]. Other assays showed that the gene transfer of p19Arf and IFN-β using our nonreplicating Ad vector in the LLC1 mouse model of lung carcinoma was able to induce markers of immunogenic cell death. In situ gene therapy with IFN-β, either alone or in combination with p19Arf, could retard tumor progression, but only the combination approach limited the progression of challenge tumors, thus acting as an in situ vaccine [59]. Thus, the p19Arf + IFN-β gene transfer approach induces oncolysis and immune activation even in the absence of viral replication, functioning as a cancer vaccine and immunotherapy, at least in mice [60,61].

We have taken great care to use different models to demonstrate the functionality of our approach since it involves IFN-β, which is known to have species-specific activities [62]. To examine our approach in human melanoma cell lines, we used the AdRGD-PG backbone to construct vectors encoding the human cDNAs p14ARF and hIFN-β and showed immunogenic cell death characterized by the emission of critical markers in vitro as well as successful ex vivo priming of human T-cells [63].

Most of the clinical trials using nonreplicating Ads are in Phase I/II (Table 1). Kumon et al. [13] have demonstrated in preclinical and clinical data the benefit of in situ Ad-REIC (adenoviral vector carrying the human *REIC*/*Dkk*-*3* gene) treatment. In preclinical data, they showed that Ad-REIC induces selective toxicity in response to endoplasmic reticulum stress and IL-7 overproduction by infected normal cells, including cells of the TME. These cells can activate innate immunity, especially NK cells, as well as cytotoxic T-lymphocytes (CTLs). In addition, DCs induced by secreted REIC proteins can present cancer antigens from apoptotic cancer cells and induce tumor-associated antigen-specific CD8^+^ CTLs. In clinical settings, preliminary outcomes have shown cytopathic effects and tumor-infiltrating lymphocytes in patients with high-risk localized prostate cancer, undergoing radical prostatectomy, who received two ultrasound-guided intratumoral injections at 2-week intervals, followed by surgery six weeks after the second injection.

Another interesting approach is gene-mediated cytotoxic immunotherapy (GMCI), which uses an adenoviral vector expressing the herpes simplex virus (HSV) thymidine kinase (TK) gene (ADV/HSV-TK), followed by an antiherpetic prodrug. The HSV-TK protein has two principal functions: (1) nucleotide analog products of prodrug phosphorylation lead to the death of dividing cancer cells, and (2) TK is a superantigen that stimulates a potent immune reaction [64]. GMCI activates the stimulator of interferon genes (STING) pathway, enhancing the production of proinflammatory cytokines such as IFNs and promoting T-cell activation. The first study using ADV/HSV-TK plus ganciclovir for the treatment of human prostate cancer was conducted by Herman et al. [65]. The patients received a single injection of the vector (10^8^ to 10^11^ vector particles) into the prostate gland in the region with the greatest concentration of tumor cells. Not only did the regimen prove safe, with minimal toxicity, but three patients that received 10^9^–10^11^ viral particles had a decrease of more than 50% in serum prostate-specific antigen (PSA) levels for periods ranging from 45 to 330 days. The safety and efficacy of GMCI to convert the TME from cold to hot have been noted for studies in different tumor types, including glioma [66], retinoblastoma [67], and mesothelioma [68].

A promising Phase III clinical trial is being conducted on patients with BCG refractory nonmuscle invasive bladder cancer (NMIBC). This disease is an early form of bladder cancer, and the recommended treatment for these patients is the use of intravesical Bacillus Calmette-Guérin (BCG). However, data has shown that around 30% to 50% of cases will recur. The outcomes of BCG-unresponsive patients are poor, and total cystectomy (complete removal of the bladder) is the standard of care for patients who are operative candidates [69].

Alternatively, patients with bladder cancer are treated with nadofaragene firadenovec (rAd–IFN-α2b/Syn3), a replication-deficient recombinant Ad carrying the interferon-α gene, which can have both TRAIL- and non-TRAIL-mediated cytotoxic effects. The patients receive the treatment directly into the bladder using a catheter every three months, and there is elevated interferon production and, consequently, increased exposure to urothelium-enhanced cytotoxic activity. Among 157 patients with carcinoma in situ, 53% of patients achieved a complete response in as early as three months, and about 24% of patients remained free of high-grade recurrence at one year. The outcomes are encouraging and currently awaiting Food and Drug Administration (FDA) approval [70].

As mentioned before, replication-defective Ads are also used as cancer vaccine strategies. GVAX, a GM-CSF gene-modified tumor vaccine, was developed by transducing autologous tumor cells with E1/E3-deleted Ad vectors encoding GM-CSF in autologous tumor cells extracted from each patient. In a phase I/II trial, 33 patients with NSCLCs that were refractory to standard treatment received the GVAX vaccine consisting of 5–100 × 10^6^ irradiated tumor cells per dose, every 2 weeks. This strategy was shown to be safe, and three patients had radiologically complete responses that lasted for more than six months [71].

Nonreplicating Ads have also been exploited as delivery vehicles in dendritic cells (DCs). Briefly, viral particles processed via proteasome result in the presentation of self and foreign antigens by MHCI and MHCII molecules to both CD8^+^ and CD4^+^ T-cells, inducing protective humoral and cellular immunity [72]. A phase I/II clinical trial tested the immune response against a vaccine consisting of autologous DCs obtained from patients, transduced ex vivo with Ads encoding the full-length melanoma antigen MART-1/Melan-A. This study pointed to an increase in CD8^+^ and CD4^+^ T-cells in 6/11 and 2/4 metastatic melanoma patients, respectively [73].

Another phase I clinical trial performed on patients with advanced NSCLC showed the induction of systemic tumor antigen-specific immune responses with enhanced CD8^+^ T-cell infiltration of tumors in 7/13 of patients. The treatment consisted of two intradermal injections of autologous DCs, transduced ex vivo with an Ad vector expressing the CCL21 gene [74]. Similar outcomes have been observed in small cell lung cancer, where 41.8% of patients presented specific anti-p53 immune responses when treated with a vaccine consisting of DCs transduced with an Ad encoding p53 [75]. These positive outcomes are not limited to solid tumors. In a phase II study, acute myeloid leukemia (AML) patients with early molecular relapse received a modified DC vaccine. The DCs were modified with two tumor-associated antigens (TAAs), survivin and MUC1, plus secretory bacterial flagellin for DC maturation and RNA interference to suppress SOCS1. The complete remission rate was 83% among all relapsed AML patients [76].

TAAs are usually expressed in normal tissues at low levels but overexpressed in tumor cells. Many TAAs have been identified as targets for tumor-reactive T-cells and can be isolated from tumor-infiltrating lymphocytes (TILs) [77]. In contrast, tumor-specific antigens (TSAs) are only encoded in cancer cells as a consequence of somatic mutations that alter the amino acid sequence, resulting in foreign proteins that can be presented to the immune system. Therefore, the neoantigens are less susceptible to the mechanisms of immunological tolerance, comprising an interesting target for vaccination [78]. Thus, gene-based vaccination using Ad vectors as a delivery agent is emerging as one of the most promising approaches for loading antigens (TAA or TSA) onto DCs. Important advantages of this modified DC approach include persistent expression of the antigen that results in activation of CD4^+^ and CD8^+^ T-cells and the induction of antibody responses and the natural adjuvant stimulating effect that Ads mediate, which contributes to DC maturation [79].

As mentioned above, ex vivo modification of DCs followed by the reintroduction of these cells in the patient is a standard strategy for these vaccines. Even so, in situ targeting of DCs has been explored using either human or murine cells, though it can often be limited by the patient’s pre-existing immunity against the adenovirus [80].

## 3. Challenges of Using Adenoviral Vectors

Even though the use of nonreplicating adenoviral vectors has shown great promise for cancer immunotherapy, several aspects of the virus and its delivery present barriers to its effectiveness. Ideally, the vector should transfer the gene to the intended cell type without causing undue antiviral host responses. In the following discussion, we present the molecular basis for Ad tropism, the anti-Ad immune responses, and the issues surrounding the systemic administration of Ad vectors. With a thorough understanding of these mechanisms, we can then explore solutions for the challenges that they pose.

### 3.1. Tissue Tropism

As mentioned previously, most human adenovirus serotypes use CAR as their primary receptor, which is expressed on several cell types, including hepatocytes, myocardiocytes, myoblasts, and epithelial and endothelial cells [51] (Table 2). Additionally, some Ads can bind to CD46, a complement regulatory protein that is present on most nucleated human cells, including hematopoietic stem cells and dendritic cells, as well as the costimulatory molecules CD80 and CD86, present on antigen-presenting cells [81,82,83]. Different primary receptors such as integrin αvβ5, heparin sulfate proteoglycans, sialic acid, and DSG2 and GD1a glycans have also been reported to support adenovirus internalization [81,84,85], with different primary receptors influencing the route of intracellular viral traffic [86].

Engineered HAd can transduce target cells and are internalized in a similar way to wild-type adenovirus infection. The internalization can be augmented by interactions between an arginine-glycine-aspartate (RGD) motif found in the penton base and integrins. Upon attaching to CAR, the fiber knob disassociates from the capsid, and the exposed penton base interacts with a secondary receptor, usually membrane integrins αvβ3 or αvβ5, responsible for virus internalization [86,88], followed by virion endocytosis via integrin-mediated signaling [86,89].

Different subgroups of adenovirus can use different types of integrins as receptors, reinforcing their characteristic cell tropism [89,90]. The virus enters the cell by a clathrin-coated vesicle and is transported in endosomes, where capsid disassembly occurs due to endosome acidity. The virion escapes the endosome and traffics to the nucleus by microtubular complexes, where replication occurs [31,85]. After 2 h, about 40% of the internalized wild-type virions arrive in the nucleus, ready to be transcribed due to their double-stranded genome. Additionally, 48–72 h after infection, nuclear and cytoplasmatic membranes are disrupted, and around 10,000 new virions are released [32].

Different serotypes may favor particular receptors; for example, HAd5 from subgroup C has been shown to utilize CAR for facilitating entry into cells [91], while HAd11 and HAd35 from subgroup B utilize CD46 as their primary receptor [83]. As mentioned, the adenoviral vectors may be chosen due to their inherent tissue-specificity and compatibility with the intended route of administration. For example, when the virus is injected into the brain, tropism for specific cell populations depends on the interaction with CAR, and lower transduction is observed with vectors that bind neither to CAR nor integrins [92]. Even so, HAd5 has been shown to enter cells by CAR-independent mechanisms, including via a hexon–lactoferrin bridge [93,94]

The native tropism of Ads for CAR on the cell surface and the interaction of viral vectors with nontarget tissues can result in toxicity and poor therapeutic efficacy. Thus, viral proteins can be genetically tailored to expand or restrict viral replication, and vector replication machinery can even be modified to augment or restrict viral replication in target cells [49]. Beyond that, nonhuman adenoviruses, such as canine (CAd2), bovine (BAd3), chimpanzee (ChAd1-7, ChAd68), and ovine (OAd7), can also be used to overcome the pre-existing immunity in human patients [95].

### 3.2. Pre-Existing Immunity in the Host

Due to the growing application of adenoviral vectors in gene therapy and vaccines, studies of seroprevalence in global populations are important. However, these studies may be limited by the lack of data from South America, Australasia, and most African countries [96]. Moreover, predominant HAdV types can change over time within a region [97], and transmission of new strains across continents appears to be frequent. A recent study conducted by Mennechet et al. [96] showed that HAdV-D26 seroprevalence appears to be relatively high in Africa and Asia and low in North America and Europe, while HAdV-B35 seroprevalence is low worldwide. HAdV-C5 is the most common serotype that infects humans, particularly in developing countries [98,99], and it is one the most commonly used adenoviral vectors; thus, the limitations on the applicability of the HadV-C5 vector, due to pre-existing immunity, have led to the construction of novel vectors derived from rare Ad serotypes [100].

HAd serotypes are often associated with specific diseases. For instance, serotypes 2–5, 7, and 21 commonly infect the respiratory tract of individuals [101,102,103,104], while serotypes 8, 19, and 37 are responsible for keratoconjunctivitis outbreaks [105,106,107]. Pharyngoconjunctivitis is often associated with serotypes 3, 4, and 7 [108,109] and acute gastroenteritis with serotypes 40 and 41 [110,111]. Likewise, neurological disorders and obesity seem to have some correlation with adenoviral infections [112,113,114]. Due to this frequent occurrence of these infections worldwide, humans have extensive preexisting immunity to adenoviruses [115,116,117,118,119].

HAd capsid proteins are very immunogenic, especially the hexon protein [120]. The host’s adaptive immunity arm detects the hypervariable regions (HVRs) in the hexon protein and releases serotype-specific neutralizing antibodies (NAs) that appear to block a postentry step [121,122]. Thus, at second contact with the same adenovirus serotype, the host NAs may rapidly neutralize it. Interestingly, at the same time, coagulation factor X (FX) in the blood binds to the hexon protein and activates complements (C4 and C4BP in the classical and alternative complement pathways) against the adenoviruses. In a fair number of tested individuals, it also protected the virions from neutralization by serum components [123]. The HAd5 vector interacts with FX, which, in turn, binds cell surface heparan sulfate proteoglycans on hepatocytes; thus, FX is essential for intravenously injected Ad5 vectors to transduce the liver [124].

Even with FX binding the hexon proteins, anti-knob fiber and anti-penton base antibodies can also prevent the adenovirus from transducing cells [125]. Nevertheless, the HAd species C knob and penton base proteins have also been shown to induce serotype-specific NAs [126].

Beyond antibodies, strong and sustained CD8^+^ T-cell responses follow adenoviral infections [127]. Up to one-third of circulating T-cells against HAd have been reported to be CD4^+^ T-cells specific for a hexon epitope conserved between HAd serotypes. Hence, the host’s preexisting CD4^+^ T-lymphocytes might promptly respond to various subsequent adenovirus serotypes in either blood or gut [128].

### 3.3. Different Administration Routes and Their Particularities

Although the administration of lower doses of Ads is well-tolerated, higher doses are known to overstimulate innate and adaptive immune responses, which might result in acute toxicity. For example, with HAd5 at the concentration 1 × 10^11^ PFU/kg, 70% of hepatocytes and 15% of Kupffer cells expressed transgene three days later [129]. Systemic delivery, which theoretically could solve the issue of reaching metastatic foci, is confounded by sequestration of the virus by the liver and the subsequent antiviral immune response as well as possible liver damage [130]. For example, at doses up to 4 × 10^12^ vp/kg of HAd5, approximately 98% of the injected virus was found in the liver 30 min after injection [131]. Thus, systemic delivery of adenoviral vectors is associated with dose-dependent toxicity and a high risk of hepatotoxicity. Several studies have shown that the delivery of adenoviral vectors to immunocompetent mice by different routes, such as intravenous [132], intraperitoneal and intratracheal [133] or via direct injection into the pancreas, resulted in the production of neutralizing antibodies, decreasing the effectiveness of a second administration [134].

The relationship between the route of administration and viral load on CD8^+^ T-cell populations has already been studied. Holst et al. [135] administered adenoviral vectors encoding β-galactosidase by intravenous or subcutaneous routes and then examined transgene-specific CD8^+^ T-cells. Independently of the route of administration, doses above 10^9^ particles were disseminated systemically. In moderate doses, both routes induced a transient peak of IFN-γ produced by CD8^+^ T-cells 2 to 3 weeks postinfection. However, with intravenous administration, these cells were only detected in the liver. Additionally, after 2 to 4 months, the systemic immunization created dysfunctional transgene-specific CD8^+^ T-cells impaired in both cytokine production and in vivo effector functions as well as the accumulation of specific CD8^+^ T-cells in the spleen. Thus, the most important influence of adenovirus administration on CD8^+^ T cell response is the route of injection and not the total antigen load [135].

In another study, the intralymphnodal administration of a nonreplicating recombinant adenoviral vector encoding the LacZ reporter gene in canine lymphosarcoma was found to be safe, with no relevant adverse effects. This finding presents the potential for its administration to lymph node metastases in both animal and human models [136]. The examples above demonstrate that efficient gene delivery using adenoviral vectors can be performed without hepatic injury or systemic immunogenicity if off-target effects are avoided. To this end, several strategies have been developed to minimize interactions of the adenoviral vector with the liver and to protect the virus from neutralizing antibodies. Some of these approaches, such as the engineering of adenovirus capsid, hexon, or fiber proteins, use of nonhuman serotypes, and nanoformulation-coated adenoviral vectors will be discussed in more detail below.

## 4. Strategies to Modify Adenovirus Tropism

Although Ads infect many different types of cells, low (or no) expression of CAR, especially in tumor cells, confounds the attachment step and represents one of the hurdles to gene therapy using adenoviral vectors. Several strategies have been employed to overcome this barrier and redirect the Ads to the intended recipient and, consequently, decrease off-target effects (Figure 1).

### 4.1. Modifications in Viral Entry: Attachment Receptors and Virus Internalization

To explore the targeting of adenovirus particles to tumor cells, initial events related to infection/transduction must be modulated: (i) viral attachment and (ii) viral entry. This strategy can target tissues by inhibiting binding to natural receptors (detargeting) in normal liver cells, for example, and, simultaneously, creating tropism for neoplasms and their metastatic foci (retargeting) [137]. Another strategy is pseudotyping, the creation of variants by recombining their capsid proteins. Here, we detail works that have used a variety of strategies to modify tropism.

The tropism of HAd5 is predominantly mediated by the interaction of fiber/knob with CAR. As an alternative to CAR-mediated viral attachment, one of the classical fiber modifications is the incorporation of the RGD sequence into the knob AB-loop motif, which greatly expands the spectrum of cell types that may be transduced [138]. Even so, the genetic incorporation of an RGD-4C peptide into the HI loop or the C-terminal end of the HAd5 fiber knob modifies the Ad knob domain without ablating native CAR-binding [139]. Another possibility is the insertion of positively charged polylysine motifs [140]. This modification permits the virus to target the tumor cell’s heparan sulfate proteoglycans, common constituents of the cell surface, and the extracellular matrix, overexpressed in several different cancer types, including cervical cancer [141].

On the other hand, directing transduction and expression of the transgenes to occur only in tumor cells (but not in normal cells) should minimize the adverse effects of the therapy. Wickham and coworkers [142] successfully used bispecific antibodies to promote the targeting of an adenoviral vector to endothelial and smooth muscle cells. However, the attempt to noncovalently associate antibodies or molecules with the surface of the viral particle may be hampered by the instability of this binding, especially if used in vivo. For this reason, the adenovirus fiber gene sequence can be edited and, thus, the peptide ligands can be incorporated directly into the protein sequence [137,143,144,145].

Taking cues from the phage display technique, adenovirus libraries can be generated with random peptide combinations and screened for their ability to transduce a particular cell type, thus refining specificity to tumor populations in a strict manner [146,147,148]. Joung et al. [148] devised a technique for producing adenovirus with modified fibers that involved cotransfecting a packaging cell with a plasmid encoding a genetically fiber-less adenovirus with a plasmid containing the open reading frames (ORFs) of the fiber of interest. Moreover, Yoshida et al. [144] developed a Cre-lox-mediated recombination system using a plasmid library encoding modified fiber and the adenoviral genome. Using this approach, these authors inserted unique peptides, each with seven random amino acids, into the AB-loop of the fiber, and, after screening, they were successful in targeting these viral vectors to glioma cells [143,149]. Although the idea seems highly promising, there are technical complications that hinder this approach. The compaction and self-assembly of the protein monomers to form the adenovirus particle is a very delicate process. The insertion of random peptides can compromise the final structure of the viral particle as well as virus production [150].

Even though two receptors are required for the adenovirus particle to penetrate the target cells, each interaction is a distinct step. While attachment receptors apparently only recognize the target cell, the HAd5 penton base–αv integrin interaction activates signaling pathways such as p38MAPK [151,152] and Rho GTPases [153], which then trigger changes in the cell cytoskeleton for endocytosis mediated by clathrin [153,154]. Interestingly, mutation of the penton base RGD sequence slows but does not impair virus internalization and infection, nor does it prevent liver tropism [86].

### 4.2. Pseudotyping the Capsid Using Components from Different Adenoviruses

Chimeric adenoviruses are usually based on HAd5 with the fiber or its knob domain replaced by that of another serotype [155]. This creates perspectives for the recombination of these subtypes and, thus, the modulation of targeting: a concept known as pseudotyping. However, the resulting range of tropisms will be restricted to the respective serotypes used in the construction and may not necessarily contemplate the range of existing receptors found in neoplasms. Table 3 summarizes important findings in studies that have employed adenovirus pseudotyping strategies.

### 4.3. Encapsulation of Adenovirus Using Synthetic Polymers

Shielding the virus with nanoparticles allows the Ad to escape immune recognition and avoid the undesirable accumulation of the vector in the liver upon systemic delivery. Furthermore, this approach can enhance the specific targeting of tumor cells. Several studies have been conducted to evaluate the efficacy of encapsulation of negatively charged Ads with cationic liposomes or particles that aim to prevent virus clearance from circulation [162].

Some Ad features, such as regular geometries, well-characterized surface properties, and nanoscale dimensions, make it a biocompatible scaffold for a wide variety of inorganic and biological structures. The Ad capsid has free lysines, the majority of them located on hexon, penton, and fiber proteins, which can be covalently linked to other molecules such as polymers, sugars, biotin, and fluorophores [163]. Polymers offer a wide range of conjugation and encapsulation that make them a safe option for immunotherapy. The main biopolymers studied are polyethylene glycol (PEG) and hydroxylpropyl methacrylamide (pHPMA), the latter being covalently bound to capsid proteins; thus, it can efficiently transduce solid tumors after intravenous injection into mice [164]. The first study using this polymer demonstrated passive tumor targeting of polymer-coated adenoviruses administered by intravenous injection; the authors observed that the coated virus accumulated inside solid subcutaneous AB22 mesothelioma tumors 40 times more than the unmodified virus [164].

As mentioned, Ad structure permits retargeting through the incorporation of synthetic molecules and antibody fragments within the virus capsid. For instance, PEG is an uncharged, hydrophilic, nonimmunogenic, synthetic linear polymer (CH_2_CH_2_O repetitions) [165] that is frequently utilized in the biopharmaceutical industry and can be useful to protect therapeutic molecules from proteolysis as well as humoral and cellular immune responses [166]. According to Fisher et al. [164], one advantage of vector PEGylation is the retention of viability after storage at various temperatures compared to conventional Ads. Covalent attachment of PEG to the adenovirus capsid may be achieved by using PEG activation mechanisms. PEG presents hydroxyl groups (OH) that make PEG-protein bonds impossible; thus, it is necessary to use chemical activation before protein attachment. In the specific case of adenoviruses, activation can be achieved through the use of tresyl-monomethoxypolyethylene glycol (TMPEG), succinimidyl succinate-monomethoxypolyethylene glycol (SSPEG), or cyanuric chloride-monomethoxypolyethylene glycol (CCPEG), which react preferentially with lysine residues in the capsid, thus supporting the formation of covalent bonds with PEG [167].

In addition to the PEGylation of the virus particle, ligands can also bind to the opposite extremity of PEG, thus providing a specific ligand to retarget the virus to the corresponding cellular receptor.

Such approaches also aid the vector in reaching distant tumor sites, as found by Eto et al. [167]. They used a cationic liposome that was composed of (1, 2-dioleoyloxypropyl)-N, N, N-trimethy-lammonium chloride:cholesterol to encapsulate the Ad vectors carrying the antiangiogenic gene (pigment epithelium-derived factor (PEDF)). The results showed that systemic administration of Ad-PEDF/liposome was well tolerated and caused the suppression of tumor growth. The coated Ad-PEDF increased apoptosis compared to uncoated Ad in the B16-F10 melanoma cell line and inhibited murine pulmonary metastasis in vivo. Moreover, Ad-luciferase encapsulated with liposome exhibited decreased liver tropism and increased transduction in the lung. Additionally, the anti-Ad IgG level after administration of the Ad-PEDF/liposome was significantly lower compared to Ad-PEDF alone. Eto et al. [167] showed that positively charged 14-nm gold nanoparticles increased the efficiency of Ad infection in mesenchymal stem cells, usually refractory to Ad transduction, mainly because CAR expression is absent or downregulated. The strategies described here support future exploration of additional formulations for liposome-encapsulated adenoviruses and their ability to target cancer cells.

### 4.4. Cancer Cell Membrane-Coated Adenoviral Vectors

Nanoparticle-based delivery systems have been extensively explored for improving cancer treatment. Cell membranes, which can be obtained from a variety of source cells, including leukocytes, platelets, red blood cells, and cancer cells, are being employed to encapsulate particles such as liposomes, polymers, silica, and Ad vectors in order to improve tumor-targeted drug delivery in addition to prolonged circulation time, reduced interaction with macrophages, and decreased nanoparticle uptake in the liver [168]. The membrane-based functions of cancer-related cells include extravasation, chemotaxis, and cancer cell adhesion [169]. As a source of cell membranes, cancer cells offer certain advantages. They can be obtained from cell lines or patient samples and possess a wide range of membrane surface proteins, such as MHC, TAAs, and neoantigens, that can program the immune system to attack local and distant tumor sites [170], as represented in Figure 2.

Tumors frequently develop a variety of mechanisms to subvert immune attack, resulting in an immune-suppressive TME. Although tumor cells can stimulate a variety of cell types, including fibroblasts, immune-inflammatory cells, and endothelial cells, through the production and secretion of stimulatory growth factors and cytokines [171], the TME can be modulated by the tumor cells themselves and tumor-infiltrating leukocytes (including regulatory T-cells (Tregs)), myeloid-derived suppressor cells (MDSCs), and alternatively activated (type 2) macrophages (M2), cytokines (IL-10, TGF-β), expression of inhibitory receptors (such as cytotoxic T-lymphocyte antigen 4 (CTLA-4) and programmed death-ligand 1 (PD-L1)) or impediment of T-cell function, resulting in the reduced effectiveness of immunotherapy [172]. Although many TAAs have been identified, their immunogenicity is generally insufficient to elicit potent antitumor responses. Typically, when the tumor reaches the malignant stage, the most immunogenic tumor-specific antigens have been eliminated via negative selection. Frequently, the nanoparticles are associated with adjuvants, secretory cytokines, antibodies, and/or viral vectors to improve the immune response [173].

Coating polymeric nanoparticles with cancer cell membranes can be used for different types of cancer therapy. For anticancer drug delivery, Zhuang et al. [173] showed that polymeric nanoparticle cores made of poly(lactic-co-glycolic acid) (PLGA), a polymer coated in an MDA-MB-435 membrane, significantly increased cellular adhesion to the source cells compared to naked nanoparticles due to a homotypic binding mechanism. For cancer immunotherapy, the authors demonstrated that a polymer coated with a B16−F10 membrane, which creates a stabilized particle, facilitated the uptake of membrane-bound tumor antigens and, consequently, the presentation and maturation of DCs. Another approach using a biohybrid (tumor-membrane-coated) nanoparticle was also able to elicit an antitumor immune response in melanoma models, changing the microenvironment profile. The administration of the vaccine enhanced the activation of APCs and increased the priming of CD8^+^ T-cells. When combining the nanovaccine with a checkpoint inhibitor, 87.5% of the animals responded, including two complete remissions, when compared to the immune checkpoint inhibitor alone. These results point to opportunities for the association of nanoparticles and immunomodulators to enhance T-cell responses [174].

The effects from the association of polymers, cancer cell membranes, and adjuvants were also observed by Fontana et al. [175]. PLGA nanoparticles were loaded with the TLR7 agonist and then coated with membranes from B16-OVA cancer cells since the presentation of foreign peptide OVA permits the tracking of responses. The nanovaccine was able to enhance uptake by antigen-presenting cells and showed efficacy in delaying tumor growth as a preventative vaccine besides displaying activity against established tumors when coadministered with the anti-programmed death 1 (PD-1) monoclonal antibody.

In another study, CpG oligodeoxynucleotide (CpG) was used as an immunological adjuvant and encapsulated into PLGA nanoparticle cores coated with membranes derived from B16–F10 mouse melanoma cells. The effect of nanoformulation on DC maturation was observed by the upregulated expression of costimulatory markers CD40, CD80, CD86, and MHC-II. Both prophylactic and therapeutic vaccines presented positive results. In the prophylactic study using the poorly immunogenic wild-type B16–F10 model, tumor occurrence was prevented in 86% of mice 150 days after challenge with tumor cells. Interestingly, mice vaccinated with the CpG-nanoformulation alone had tumor growth comparable to the control group and a median survival of 22 days. This reinforces the role of the cancer cell membrane in targeting the elimination of malignant cells by the immune system. In the therapeutic model, mice challenged with B16–F10 cells and subsequently treated with the nanoformulation presented a modest ability to control tumor growth. However, the combination of nanoformulation and a checkpoint blockade cocktail (anti-CTLA4 and anti-PD1) significantly enhanced tumor growth control. As such, the results encourage further research into nanoparticle vaccine formulation and possible associations with other immunotherapies that modulate different aspects of immunity [176].

In a different strategy, Fusciello et al. [177] combined an oncolytic virus (due to its natural adjuvant properties) and cancer cell membranes carrying tumor antigens. They found that viral transduction was significantly increased with the coated virus, implying an uptake mechanism different than that utilized by the naked virus, which requires CAR, representing a significant advantage for transducing CAR-negative cell lines. Additionally, the coated virus was better able to control tumor growth compared to other treatments. The vaccination using coated viruses created a highly specific anticancer immune response, redirecting the immune response against the tumor [177]. Thus, personalized cancer vaccines can represent an alternative approach to target cancer even without determining specific antigens for each patient. We hypothesize that this approach will also be applicable to nonreplicating adenoviral vectors, though this has not yet been shown.

### 4.5. Association of Antibodies and Viral Structures

The incorporation of antibodies into the viral structure is another interesting option for creating specificity. Despite the obstacles to the use of conventional antibodies (human, murine, and goat), smaller molecules from other species, such as alpacas, can be added to the structure of the Ad capsid without disturbing its synthesis and assembly. For example, van Erp et al. [178] generated a single domain camelid antibody against the human carcinoembryonic antigen present in human colorectal adenocarcinoma cells. They incorporated this molecule into the adenovirus capsid, achieving a more specific tropism for tumor cells and reducing off-target toxicity. Although the strategy was developed to retarget oncolytic viruses, it can also be used to improve nonreplicative adenoviral vectors.

Despite the cited possibilities, the need to re-engineer vectors de novo for each novel target may be an unnecessary and costly effort. Since it is possible to combat different kinds of cancers through similar molecular mechanisms, such as the induction of immunogenic cell death, the development of adaptable platforms may allow the establishment of virus-based therapies in a more scalable and affordable way. Such approaches may, in the future, permit low-effort adaptation of pre-existing therapies to target different cellular markers and treat other tumors.

A viable alternative may be the use of adapter molecules. Bhatia et al. [179] developed an anti-CXCR4 bispecific adapter (sCAR-CXCL12). Chemokine receptor type 4 (CXCR4) is known to be overexpressed in a wide variety of cancers, such as melanoma [180] and breast cancer [181], and it is associated with metastasis and poor overall survival. Bhatia et al. [179] designed a recombinant adapter molecule composed of an ectodomain portion of the human CAR, followed by a 5-peptide linker (GGPGS) and a 6-His tag sequence, fused to the mature human chemokine CXCL12/SDF-1a sequence (CXCR4 ligand). According to the researchers, this bispecific adapter attenuated liver infection in vivo and a promoted a considerable increase in cancer cell infection, as observed in xenograft tumors in mice.

In another interesting work, Schmid et al. [182] achieved, simultaneously, the retargeting of type 5 adenovirus tropism to a specific cancer marker and the reduction of its liver sequestration. Unlike other adapter strategies, they utilized designed ankyrin repeat proteins (DARPins). Similar to antibodies, these proteins can bind to a target with rather good specificity. Moreover, these molecules can be engineered to target different antigens on the cell surface [183]. Schmid et al. [182] designed an adenovirus-antigen adapter composed of three monomers. Each monomer was made of a retargeting DARPin, a flexible linker, a knob-binding DARPin, and a trimerization motif [182,184]. The last component is responsible for the stability of the complex, allowing the coating of the adenovirus fiber knob and, consequently, impeding virus natural tropism. In addition, according to the researchers and some early works, the removal of CAR and integrin interactions may reduce liver tropism [185,186], an effect also observed when those capsid sites are blocked by DARPin adapters. Furthermore, this protein was able to hide the region responsible for adenovirus–liver interaction without disturbing the adenovirus–integrin interaction. Nonetheless, the researchers developed an adenovirus-binding molecule, named “shield”, derived from humanized antihexon scFv, which was designed to bind to hexon proteins, effectively protecting them from neutralizing antibodies [182].

### 4.6. Genetic and Chemical Capsid Modifications and Association with Polymers

Other strategies have emerged that support the retargeting and detargeting of ade-noviral vectors. For example, the CGKRK peptide mediates the targeting of tumor cells and tumor neovasculature and has been tested for its ability to retarget PEGylated adenoviral vectors: PEG molecules are conjugated to the surface of the viral vector; then, the peptide is attached via a chemical reaction, resulting in its conjugation to the functional group of PEG [187]. Moreover, Bonsted and colleagues demonstrated that a linker between the poly(2-(dimethylamino)ethyl methacrylate) (pDMAEMA) and the epidermal growth factor (EGF), commonly overexpressed in tumors, efficiently transduced CAR-deficient cells [188,189]. Additionally, an EGF mimetic peptide linked to the cationic PAMAM (polyamidoamine) dendrimer polymer through a PEG linker has been used to retarget dendrimer-coated Ad vectors; it has been shown to increase transgene expression in target cells compared with the untargeted vector [190].

Kreppel et al. [191] introduced a genetic–chemical concept for vector re- and detargeting. For that, the authors genetically modified the virus in order to present cysteine residues in the capsid, including the fiber HI-loop [191], protein IX [192], and hexon [193]. The cysteine residues were then covalently modified with thiol-reactive coupling moieties, including ligands, shielding polymers, carbohydrates, small molecules, and fluorescent dyes [194]. Kreppel et al. demonstrated that amine-based PEGylation and thiol-based coupling of transferrin to the fiber knob HI-loop successfully retargeted the modified Ad vectors to CAR-deficient cells [191].

These studies highlight the possibility of creating adenoviral vector platforms that need no further genetic modification; thus, a wide variety of target tissues may be explored with the aim of improving specificity and decreasing the neutralizing effects of preexisting antibodies.

## 5. Conclusions and Future Perspectives

Many features make Ads interesting vehicles for the delivery of foreign antigenic proteins or gene therapy: large cloning capacity, genetic stability, and high in vivo transduction capacity in both dividing and nondividing cells. The natural antiviral immune response can be useful to reprogram the tumor microenvironment from “cold” to “hot” by inducing T-cell-specific immune responses and proinflammatory cytokine expression [195]. The success of therapy depends on several other factors, such as the quality, intensity, specificity, and half-life of immune responses against the tumor. In this scenario, neoantigens have emerged as an attractive target for cancer therapy. Major advances in using the non-self-peptides are the absence of pre-existing central tolerance, potential strong immunogenicity, and lower risk of autoimmunity diseases [196]. We expect that continued refinement of Ad vector design and a deeper understanding of neoantigens will converge to provide an exceptional platform for cancer immunotherapy.

Even so, we point out some limitations for the use of neoantigens in personalized medicine: (1) neoantigens are limited by the diversity of somatic mutations in different tumor types and their individual specificity; (2) the probability that the neoantigens are shared between patients is very low; (3) identification and verification of neoantigens is still time-consuming and expensive [197]. In addition, the construction of adenoviral vectors encoding each neoantigen would be costly and time-consuming; thus, approaches that do not require vector construction may be preferable, including the use of peptides and membrane coatings.

Otherwise, the effectiveness in the use of neoantigens has already been observed in preclinical [198,199] and clinical data [200,201,202]. In addition, patients with high mutation burden tumors, like melanoma [203,204], non small-cell lung cancer [205], and bladder cancer [206], have had more clinical benefit from checkpoint-blockade therapy than those with lower mutation loads [196]. Moreover, the prediction of peptides binding to MHC molecules and, consequently, the identification of neoepitopes able to stimulate the immune response are emerging as novel approaches that could be associated with adenoviral vectors, reversing part of the tumor-induced immunosuppression.

Recently, D’Alise and collaborators [207] demonstrated the satisfactory benefits of genetic vaccines based on Ads derived from nonhuman great apes (GAd) encoding multiple neoantigens applied in the CT26 murine colon carcinoma model. Both prophylactic and early therapeutic vaccinations elicited strong and effective T-cell responses and controlled tumor growth in mice. The tumor-infiltrating T-cells were diversified in animals treated with GAd and anti-PD1 compared to anti-PD1 alone [207]. The big challenge of neoantigens is the complexity in identifying immunogenic antigens unique to each patient. However, more optimized sequencing platforms and bioinformatics tools are helping to make personalized therapy truly viable. All in all, the data presented here highlight new perspectives of cancer vaccines and gene therapy using modified nonreplicating adenoviruses and different strategies to turn the immune response against the tumor more specific and robust, contributing to local and distant control of tumor progression.

Although viral delivery systems are quite promising strategies in gene therapy, there are some limitations to their clinical application. The major barriers are host immune responses that result in the clearance of vectors, interaction with plasma proteins, liver sequestration, Ad CAR-dependence, and off-target effects [208]. Regarding these issues, a number of genetic manipulations have been exploited to redirect adenovirus binding to different cell surface receptors and, consequently, increase affinity for the target, with lower adverse effects [50].

In this scenario, different strategies using coated viruses have emerged in recent years, and both biological and chemical approaches can be used to coat the virus and improve delivery, especially for the systemic route. Since these strategies involve using cancer cell membranes that can be obtained directly from tumor cell lines, they provide greater biocompatibility with the tumor site and, consequently, specifically target these cells [209]. A growing body of evidence suggests that cancer cell membrane-coated viruses can be delivered by the systemic route, improving the targeting of metastases, with higher retention time, lower immune recognition, and decreased liver sequestration, toxicity, and accumulation in healthy tissues. The induction of immunogenic cell death by nonreplicating Ad vectors is associated with innate immune responses, antigen processing and presentation, and, finally, the activation of the cellular immune response. While few examples currently exist of using membrane-coated adenoviral vectors, we hypothesize that this approach will continue to be studied, including in nonreplicating vectors.

In summary, improvements in vector delivery and targeting will provide an even greater potential for the use of nonreplicating adenoviral vectors in cancer immunotherapy. In particular, we envision vectors adapted to support systemic delivery, achieve tumor specificity, induce tumor cell death and supply specific antigens to guide antitumor immune responses.

## Figures and Tables

**Figure 1 cancers-13-01863-f001:**
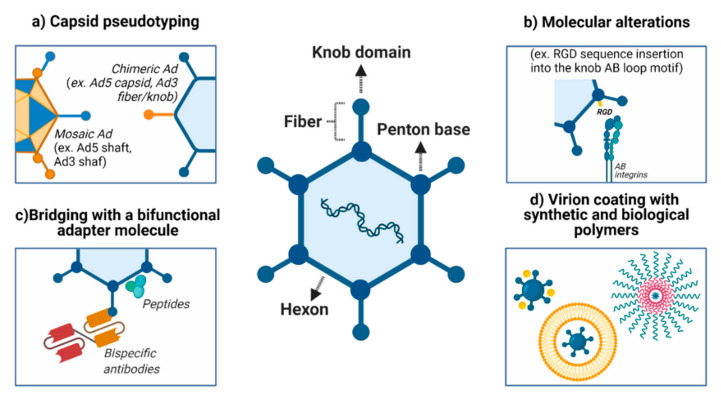
Improvements in vector delivery to and targeting of cancer cells. (**a**) Several approaches can be used to modify viral attachment and entry, such as inhibiting the binding to natural receptors (detargeting) and creating tropism for neoplasms and their metastatic foci (retargeting). (**b**) An alternative to CAR-mediated viral attachment is modifying the fiber (for example, incorporation of the RGD sequence into the knob AB-loop motif). (**c**) Ad structure permits retargeting through the incorporation of synthetic molecules and antibody fragments within the virus capsid. (**d**) Both biological (e.g., cell membrane, liposome) and chemical (e.g., gold, silver, PEG) approaches may be used to coat the virus and improved delivery, especially for the systemic route. PEG: polyethylene glycol. Created with BioRender.com.

**Figure 2 cancers-13-01863-f002:**
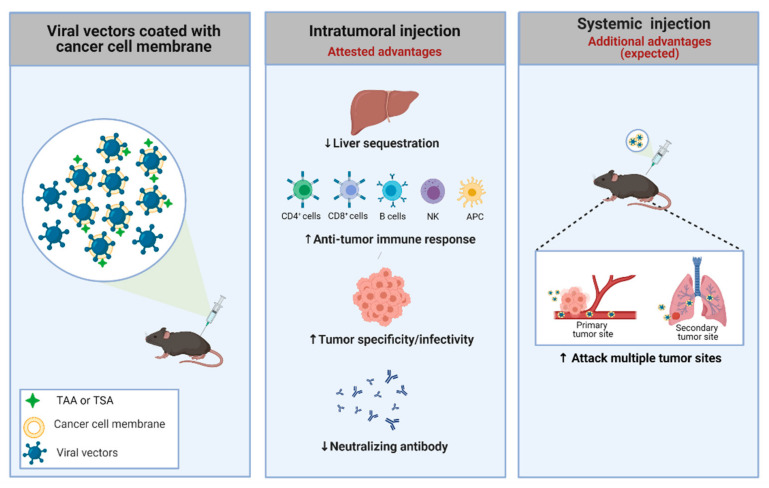
Strategy used to improve the specificity of adenoviral vectors. An adenovirus coated with a cancer cell membrane has some advantages, such as the presence of TSA and TAA, which aids the anti-tumor immune response. Additionally, the membrane can be engineered to present specific molecules/receptors, improving the power of interaction with the tumor. Moreover, the viral coating can offer several benefits, including suppression of liver toxicity, increase of specific infectivity to cancer cells, preferential antitumor (not antiviral) immune response, and escape from pre-existing neutralizing antibodies in both routes of delivery (intratumoral and systemic). Systemic administration using virus coated with membranes could offer a highly desirable outcome: targeting metastatic foci. TAA: tumor-associated antigens; TSA: tumor-specific antigens. Created with BioRender.com.

**Table 1 cancers-13-01863-t001:** Clinical trials using nonreplicating adenoviral vectors for cancer gene therapy.

Vector	Transgene	Cancer	Mechanism	Therapy	Phase	Clinical Trial/ Reference	Status
Ad5-SGE-REIC/Dkk3	REIC/Dkk3	Localized prostate cancer	Cancer cell death induction and anticancer immunity	Neoadjuvant	I/II	NCT01931046 [13] #	Active, not recruiting
Ad5-SGE-REIC/Dkk3 (MTG201)	REIC/Dkk3	Relapsed malignant pleural mesothelioma	Cancer cell death induction and anticancer immunity	Combination with nivolumab	II	NCT04013334 [14] #	Active, recruiting
AdHSV-tk /GCV	HSV-tk Ad-hCMV-Flt3L	High-grade malignant gliomas	TK: direct tumor cell killing Flt3L: immunostimulating effects		I/II	NCT01811992 [15] #	Active, not recruiting
Adv/tk (GMCI)	HSV-tk	Advanced nonmetastatic pancreatic adenocarcinoma	TK: direct tumor cell killing	Neoadjuvant plus chemoradiation	II	NCT02446093	Active, not recruiting
Adv/tk	HSV-tk	Advanced hepatocellular carcinoma	TK: direct tumor cell killing	Liver transplantation	III	NCT03313596 [16] #	Active, recruiting
Adv/tk (GMCI)	Adv-tk	Pediatric brain tumors	Direct tumor cell killing	Combination with radiation therapy	I	NCT00634231 [17] #	Active, not recruiting
Adv/RSV-tk	HSV-tk	Recurrent prostate cancer	Direct tumor cell killing	Combination with brachytherapy	I/II	NCT01913106	Active, recruiting
Adv/HSV-tk	HSV-tk	Metastatic nonsmall cell lung carcinoma and uveal melanoma	Direct tumor cell killing	Combination with stereotactic body radiation therapy or nivolumab	II	NCT02831933	Terminated (Lack of funding)
Ad/PNP + fludarabine	PNP	Head and neck squamous cell carcinoma	PNP protein actives the second component of the therapy fludarabine phosphate		I	NCT01310179 [18]	Completed
rAd-IFN/Syn-3 (Instiladrin)	IFN α-2b	High-grade nonmuscle invasive bladder cancer	Immunoregulatory effects		III	NCT02773849 [19] #	Active, not recruiting
BG00001	IFN-β	Pleural mesothelioma	Immunoregulatory effects		I	NCT00299962 [20]	Completed
Ad-RTS-hIL-12	IL-12	Advanced or metastatic breast cancer	Proinflammatory cytokine, enhances the cytotoxic activity of T-lymphocytes and resting natural killer cells	Combination with VELEDIMEX	Ib/II	NCT02423902	Unknown
Ad-RTS-hIL-12	IL-12	Recurrent or progressive glioblastoma	Proinflammatory cytokine, enhances the cytotoxic activity of T-lymphocytes and resting natural killer cells	Ad-RTS-hIL-12 + Veledimex in combination with Cemiplimab	II	NCT04006119	Active, not recruiting
Ad-RTS-hIL-12	IL-12	Glioblastoma or malignant glioma	Proinflammatory cytokine, enhances the cytotoxic activity of T-lymphocytes and resting natural killer cells	Combination with Veledimex	I	NCT02026271 [21] #	Active, not recruiting
SCH-58500	P53	Primary ovarian, fallopian tube, or peritoneal cancer	Tumor suppressor gene: antitumor effect by blocking cell cycle progression at the G1/S, activating DNA repair pathways		I	NCT00002960 [22]	Completed
Ad-p53	P53	Recurrent or metastatic head and neck squamous cell carcinoma	Tumor suppressor gene: antitumor effect by blocking cell cycle progression at the G1/S, activating DNA repair pathways	Adjuvant in combination with Anti-PD-1 or Anti-PD-L1 therapy	II	NCT03544723	Active, recruiting
ADVEXIN	P53	Squamous cell carcinoma of the oral cavity, oropharynx, hypopharynx, and larynx	Tumor suppressor gene: antitumor effect by blocking cell cycle progression at the G1/S, activating DNA repair pathways		I/II	NCT00064103 [23]	Completed

# partial outcomes.

**Table 2 cancers-13-01863-t002:** Classification and tropism of human adenoviruses.

Classification and Tropism of Human Adenoviruses			
Subgroup	Serotypes	Identified Receptors	Tropism
A	12, 18, 31, 61	CAR	Enteric, respiratory
B	3, 7, 11, 14, 16, 21, 34, 35, 50, 55, 66,68, 76–79	CD46, DSG2, CD80, CD86	Renal, ocular, respiratory
C	1, 2, 5, 6, 57, 89	CAR, VCAM-1, HSPG, MHC1-a2, SR	Ocular, lymphoid, respiratory, hepatic
D	8–10, 13, 15, 17, 19, 20, 22–30, 32, 33, 36–39, 42–49, 51, 53, 54, 56, 58–60, 62–65, 67, 69–75, 80–88, 90–103	SA, CD46, CAR, GD1a	Ocular, enteric
E	4	CAR	Ocular, respiratory
F	40, 41	CAR	Enteric
G	52	CAR, AS	Enteric

CAR: coxsackie adenovirus receptor. DSG2: desmoglein-2. GD1a: GD1a ganglioside. HSPG, heparin sulfate proteoglycans. MHC1-a2: major histocompatibility complex-a2. SA: sialic acid. SR: scavenger receptor. VCAM-1, vascular cell adhesion molecule-1. Adapted from [85,87].

**Table 3 cancers-13-01863-t003:** Different studies using adenovirus pseudotyping strategies.

Attachment Receptor	Tropism	Modification	Serotype Origin/ Subgroup	Results	Reference
CD46	Glioma	Fiber replacement	Ad35, Ad16, Ad50	Increased transduction of patient-derived cells	[156]
Adenovirus serotype 3 receptor	Ovarian cancer cells	Fiber knob replacement	Ad3 (modified)/B1	Enhanced gene transfer to various cancer cell lines and primary tumor tissues	[157]
Adenovirus serotype 3 receptor	Lung cancer (NSCLC primary tissue)	Fiber knob replacement	Ad3 (modified)/B1	Improved killing of NSCLC cells	[158]
Sialic acid, phage display for kidney	Renal cancer and detargeting the liver	Fiber knob replacement	Ad5 (modified)/19p (fiber)	Reduced liver tropism and improved gene transfer to renal cancer	[159]
Unidentified cellular receptor	Cancer cell lines of pancreatic, breast, lung, esophageal, and ovarian	Fiber knob replacement	Ad5 (modified)/D49	Efficiently transduced	[160]
CD46	Primary human cell cultures	Fiber replacement	Ad5PTD/F35	Increased transduction capacity of T-cells, monocytes, macrophages, dendritic cells, pancreatic islets, mesenchymal stem cells, and tumor-initiating cells	[161]

PTD: Tat-PTD hexon modification.

## Data Availability

No new data were created or analyzed in this study. Data sharing is not applicable to this article.

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
