# Peer review of "Nonreplicating Adenoviral Vectors: Improving Tropism and Delivery of Cancer Gene Therapy"

_cancers, 2021, doi:10.3390/cancers13081863_

Round 1

Reviewer 1 Report

The review article “Non-replicating adenoviral vectors: Improving tropism and delivery for cancer gene therapy” by Tessarollo et al. presents a well written overview on several aspects of (non-oncolytic) AdV gene therapy in cancer.

The authors largely have a good track record of AdV work and several chapters are written with a noticeable authority. The tables and figures are of good quality and support the narrative of the authors.

There are, however, some conceptual aspects that could be improved in my opinion. There is also a considerable number of minor errors that are listed below.

The authors set out to make a case for the promise of non-replicating AdV in cancer therapy. However, in my opinion, the distinction between oncolytic and non-replicating AdV is too vague. Throughout the manuscript several references are used to support statements or concepts which so far have only clearly been shown with oncolytic viruses. While such conjecture could maybe hold for some cases, it misses the point if the main intention of the authors is to highlight the useability of NON-REPLICATING AdV, and even stating that they might potentially be superior (lines 57-58). Such examples where OV applications where used are for instance the discussion of the positive effect of antiviral immunity (ref Gujar et al, #167) or reversal of immune-inhibitory tumor microenvironment. While I like Table 1 which shows that replication incompetent AdV gene transfer vectors are clinically tested, such clear distinction is missing in some chapters that discuss novel aspects or potential future applications. A table that clearly comprehensively lists examples of non-replicating AdV studies for the respective concept (cancer vaccine, TME remodeling, immune-modulatory cargos etc).

While I like the chapter 3.2. and this is clearly a very important topic, I do believe the induction of antivector immunity differs between non-replicating and oncolytic AdV. I would assume this might have been addressed by some studies in AdV permissive hosts.

Chapter 4.4 (and Fig 2)

The authors claim that cancer cell membrane coated non-replicating AdV can serve as some sort of vaccine by exposing membrane bound TAAs or TSAs. Of all the studies they cite, only 157 shows experimental data linking AdV with such a hypothesis. However, as the authors correctly state, here an oncolytic AdV is used. The other references listed here my opinion are conjecture based on nanoparticle studies. Unless the authors could clearly show that such a principle has clearly been confirmed with cell membrane coated replication incompetent AdV, this section should be revised and presented as a hypothesis.

Lines 571-577: although the authors just recite findings from reference 155, this reference is flawed in my opinion. OVA is NOT a membrane bound protein! Ova-expressing B16 cells present only OVA peptide fragments via their MHC class 1 molecules. The narrative of these membranes containing actual tumor associated antigens (OVA) is in my opinion not the correct expression.

Minor:

Line 56: delete: “to applicable”

Line 61-63: Is this sentence an opinion or can the authors provide confirmation that viral replication might not be of advantage in a vaccine setting (please note that many preclinical immune studies are performed in mice which do not support AdV replication).

Line 66: delete 2nd “that”

Line 68: “propose” instead of “to propose”

Line 78: plural: hexons

Line 86: virus entry. The phrasing is somewhat misleading. The “internalization” does not “activate” clathrin endocytosis. Internalization is mediated via clathrin endocytosis.

Line 107: “hence prevent”

Line 203: “prove”

Line 267: wording: “exposed” might not be the best word in this context

I like Table 2 a lot but I wonder if in the text some general ranking could be included on which are the serotypes that more frequent or less frequent show seropositivity in the population (pre-existing immunity).

While it is good to distinguish ex vivo and in vivo gene transfer, I think it would be better to more clearly highlight this fundamental seperation earlier than page 8.

Line 360-369, please specify low and high concentration ranges for the non-AdV specialized reader

Line 374 please make sure the final manuscript correctly indicates exponential values (here 10^9 instead of 109)

Line 385: language / grammar: double plural and metastasES

Line 385-390: unless it has clearly been shown to relate to ocular CANCEROUS disease (which I doubt), the eye injection route is probably not relevant for this manuscript discussing AdV gene therapy for cancer. General gene therapy applications of AdV that have no bearing for an oncology application are probably somewhat distracting

Line 426: please correct: “(113) and coworkers…”

Table 3:

It is the opinion of this reviewer, that this extensive manuscript should not unnecessarily be conflated with non-cancer related AdV gene therapy aspects. I believe table 3 could be improved by focusing on cancer applications. For instance, listing retargeting approaches for cardiovascular tissue, smooth muscle, normal airway epithelia is distracting in this context.

Lines 511-512, grammar

Line 521: grammar

Figures: if scientific illustration tools such as “biorender” has been used to generate the figures (is is obvious in figure 2), such use of the software should be indicated.

Line 594: please check references. Kroll at all is 156 not 157, conversely, the Kroll paper does not talk about AdV OVs, the Fusciello et al paper does (ref 157)

General note: it appears that the numerical reference list and the references in the text are not correctly linked. Line 608/609 ref Fusciello is 157, not 158. Line 621 ref Erp is 158 not 159 etc….

634: please correct: “In also,….”

Author Response

Tessarollo et al
Point-by-point response to Reviewers
#Reviewer 1:
Comments and Suggestions for Authors
The authors set out to make a case for the promise of non-replicating AdV in cancer therapy. However, in my opinion, the distinction between oncolytic and non-replicating AdV is too vague. Throughout the manuscript several references are used to support statements or concepts which so far have only clearly been shown with oncolytic viruses. While such conjecture could maybe hold for some cases, it misses the point if the main intention of the authors is to highlight the useability of NON-REPLICATING AdV, and even stating that they might potentially be superior (lines 57-58). Such examples where OV applications where used are for instance the discussion of the positive effect of antiviral immunity (ref Gujar et al, #167) or reversal of immune-inhibitory tumor microenvironment. While I like Table 1 which shows that replication incompetent AdV gene transfer vectors are clinically tested, such clear distinction is missing in some chapters that discuss novel aspects or potential future applications. A table that clearly comprehensively lists examples of non-replicating AdV studies for the respective concept (cancer vaccine, TME remodeling, immune-modulatory cargos etc).
We appreciate the Reviewer’s suggestion. The paragraph starting at line 56 has been modified in order to maintain focus on the non-replicating vectors. Table 1 has been updated in order to reflect the variety of approaches that use non-replicating adenoviral vectors for different aspects of cancer therapy.
While I like the chapter 3.2. and this is clearly a very important topic, I do believe the induction of antivector immunity differs between non-replicating and oncolytic AdV. I would assume this might have been addressed by some studies in AdV permissive hosts.
Additional information was added (paragraph starting on line 158) showing some differences in immune response when using non-replicating vs. replicating AdV.
Chapter 4.4 (and Fig 2): The authors claim that cancer cell membrane coated non-replicating AdV can serve as some sort of vaccine by exposing membrane bound TAAs or TSAs. Of all the studies they cite, only 157 shows experimental data linking AdV with such a hypothesis. However, as the authors correctly state, here an oncolytic AdV is used. The other references listed here my opinion are conjecture based on nanoparticle studies. Unless the authors could clearly show that such a principle has clearly been confirmed with cell membrane coated replication incompetent AdV, this section should be revised and presented as a hypothesis.
The Reviewer is correct, to date we know of only one example of membrane coated Ad, and this being a replicating vector. We have altered Chapter 4.4 and the conclusion to better reflect this point.
Lines 571-577: although the authors just recite findings from reference 155, this reference is flawed in my opinion. OVA is NOT a membrane bound protein! Ova-expressing B16 cells present only OVA peptide fragments via their MHC class 1 molecules. The narrative of these membranes containing actual tumor associated antigens (OVA) is in my opinion not the correct expression.
Indeed, the use of cellular membranes may also provide peptides associated with MHC. We adjusted the text (lines 586 and 631) to reflect this point. We appreciate this correction.
Minor:
-Line 56: delete: “to applicable” Done.
-Line 61-63: Is this sentence an opinion or can the authors provide confirmation that viral replication might not be of advantage in a vaccine setting (please note that many preclinical immune studies are performed in mice which do not support AdV replication).
This phrase has been eliminated.
-Line 66: delete 2nd “that” Done.
-Line 68: “propose” instead of “to propose” Done.
-Line 78: plural: hexons Done.
-Line 86: virus entry. The phrasing is somewhat misleading. The “internalization” does not “activate” clathrin endocytosis. Internalization is mediated via clathrin endocytosis.
We apologize for this mistake. We have now altered the text with a summary of viral entry (lines 85-91) and also a brief description of additional mechanisms in section 3.1.
-Line 107: “hence prevent” Done.
-Line 203: “prove” Done.
Line 267: wording: “exposed” might not be the best word in this context
We agree and have changed the word “exposed” to “mentioned”.
-I like Table 2 a lot but I wonder if in the text some general ranking could be included on which are the serotypes that more frequent or less frequent show seropositivity in the population (pre-existing immunity).
As suggested, additional information has been added in section 3.2 (Lines 342-350).
While it is good to distinguish ex vivo and in vivo gene transfer, I think it would be better to more clearly highlight this fundamental seperation earlier than page 8.
These routes of administration are now defined at the start of section 2. We appreciate this suggestion (lines 121-127).
Line 360-369, please specify low and high concentration ranges for the non-AdV specialized reader
Practical examples of the impact of virus concentration have been added to section 3.3.
Line 374 please make sure the final manuscript correctly indicates exponential values (here 10^9 instead of 109) Done.
Line 385: language /grammar: double plural and metastasES Done
Line 385-390: unless it has clearly been shown to relate to ocular CANCEROUS disease (which I doubt), the eye injection route is probably not relevant for this manuscript discussing AdV gene therapy for cancer. General gene therapy applications of AdV that have no bearing for an oncology application are probably somewhat distracting.
The text and reference related to ocular gene transfer was removed.
Line 426: please correct: “(113) and coworkers…” Done.
Table 3: It is the opinion of this reviewer, that this extensive manuscript should not unnecessarily be conflated with non-cancer related AdV gene therapy aspects. I believe table 3 could be improved by focusing on cancer applications. For instance, listing retargeting approaches for cardiovascular tissue, smooth muscle, normal airway epithelia is distracting in this context.
We agree. Non-oncologic targets were removed and additional studies in cancer have now been added.
Lines 511-512, grammar
Moreover, Ad-luciferase encapsulated with liposome exhibited a decreased liver tropism and increase the in lung.
This phrase has been corrected.
-Line 521: grammar
This phrase has been corrected.
Figures: if scientific illustration tools such as “biorender” has been used to generate the figures (is is obvious in figure 2), such use of the software should be indicated.
We have added this information. The end of each legend now indicates: “Created with BioRender.com”
Line 594: please check references. Kroll at all is 156 not 157, conversely, the Kroll paper does not talk about AdV OVs, the Fusciello et al paper does (ref 157). Done.
General note: it appears that the numerical reference list and the references in the text are not correctly linked. Line 608/609 ref Fusciello is 157, not 158. Line 621 ref Erp is 158 not 159 etc….
Thank you for the attention to detail. We have checked all references and modified the numerical citations.
634: please correct: “In also,….” Done

Reviewer 2 Report

This is an extensive ambitious review presenting  advantages and  several obstacles  that have to be mastered  during development of  non replicating adenovirus vectors for use the  treatment of cancer.

Line 95 typo virons

 line 172 to which extent  would experimental work in the Ad5 immune mice mimic  the human situation

Would a human adenovirus with  very low seroprevalence such as Ad26  that can interact  with   CD46  via all hexons  of the capsid  be a better alternative? (D Persson et al PNAS  2021)

Ref 12 refers to  peptide VI  Urs Greber has described its role inthe following way The capsid is mechanically  torn apart then  the amphipatic peptide VI  will insert into the cytoplasm membrane  thereby causing breaches  allowing calcium  to flush into the cell.    This will activate SAMse  within the lysosomes thus sfingomyelin will be cleaved into phosphorylcholine  and   ceramide.  Large amounts of ceramide will facilitate   the passage  of the adenovirus into and thorough  the   host celll.

line 374 doses above 109  or a billion?

line  385 typo metastasi

line 421  typo  tropismo

line 456 70 serotypes  of human adenoviruses is not correct.

Table 3 Receptor X  has four  references  What does  X represent?

line 466  the receptor for Ad19a/64  is labled not determined  -yet   this  virus  and Ad37    uses  sialic acid   as their  receptor.

How are  polymer coated  adenovirusvectors uncoated?

line 542  spell out TME

Describe the G Advaccine.

Author Response

Tessarollo et al
Point-by-point response to Reviewers
#Reviewer 2: Comments and Suggestions for Authors This is an extensive ambitious review presenting advantages and several obstacles that have to be mastered during development of non replicating adenovirus vectors for use the treatment of cancer. -Line 95 typo virons Done -line 172 to which extent would experimental work in the Ad5 immune mice mimic the human situation We found this question to be very pertinent. In our own work we have taken great care to examine IFN-β activity in both mouse and human models of melanoma. So far, we have seen similar responses in both systems. The text has been modified to reflect this point (lines 206 to 212). We also recognize the differences between mouse and human immune systems and the possible impact in translating results obtained in mice to the human, clinical situation. Genomic comparisons of mice and humans revealed significant overlap in transcriptional programs, but also exposed noteworthy differences (Tao et al 2017). Atasheva et al 2019 also highlight that there is no animal model system where HAdv would productively replicate at the same level observed in susceptible human cells. However, mice are still the foundation of basic research. Studies in mice provide an advance in understanding of mechanisms of virus recognition by the innate immunes system and important inferences can be made using the murine model. -Would a human adenovirus with very low seroprevalence such as Ad26 that can interact with CD46 via all hexons of the capsid be a better alternative? (D Persson et al PNAS 2021) Thank you for this comment. HAdv-C5 is a well-characterized virus and decades of studies have provided in-depth knowledge and confidence in the use of this vector despite some limitations in clinical application (Padilla et al 2016). Other studies led to the discovery of less prevalent adenoviruses, such as HAdv-26 and HAdv-35. The low prevalence may be an opportunity to avoid neutralizing antibodies and innate immune response if these serotypes are used as vectors. A point that deserves attention is about the ability of these adenoviral vaccine vectors to induce an immune response. Chen et al 2010 had demonstrated that the immune response promoted to HAdV-C5 is more potent. The work by Persson et al shows that interaction with CD46 occurs exclusively through the hexon protein. Recombinant Ad26 vectors have been shown to bypass Ad5 neutralizing antibodies (Geisbert et al, 2011). Recent events have shown that Ad26 can be used successfully as a vaccine platform, in particular the J&J vaccine for SARS-CoV-2. Such vectors are also being developed for Zika and Ebola virus vaccines. So, yes, the Ad26 platform may provide advantages, especially in terms of escaping pre-existing, Ad5 neutralizing antibodies. Use of Ad26 for anti-cancer approaches is not widely explored, though some examples from the lab of Michael Barry do point to their utility for intra-tumoral gene transfer or for a vaccine approach.
-Ref 12 refers to peptide VI Urs Greber has described its role inthe following way The capsid is mechanically torn apart then the amphipatic peptide VI will insert into the cytoplasm membrane thereby causing breaches allowing calcium to flush into the cell. This will activate SAMse within the lysosomes thus sfingomyelin will be cleaved into phosphorylcholine and ceramide. Large amounts of ceramide will facilitate the passage of the adenovirus into and thorough the host cell. We really appreciated this observation. We have updated the description of adenovirus entry into cells since this was repeated in different sections. In the current form, we chose to summarize these mechanisms, thus the reference by Greber et al has been removed. Updated text found in: “Overview: Structural and molecular features of non-replicating adenoviral vectors” (Lines 85-90). -line 374 doses above 109 or a billion? The exponent is now correct in the text (10^9). -line 385 typo metastasi Done. -line 421 typo tropismo Done. -line 456 70 serotypes of human adenoviruses is not correct. We apologize, this mistake has been removed from the text. -Table 3 Receptor X has four references What does X represent? We apologize for this oversight. The receptor X represents the adenovirus serotype 3 receptor. We have corrected this point in the table. -line 466 the receptor for Ad19a/64 is labled not determined -yet this virus and Ad37 uses sialic acid as their receptor Thank you for this comment. This issue has been resolved since we excluded all studies not related to cancer. How are polymer coated adenovirusvectors uncoated? While this is certainly an important topic, we have not addressed polymer uncoating. This may vary depending on the polymer in question and could potentially impact cellular response, but we feel that such a discussion is beyond the scope of this review where we have focused on targeting strategies and how they may impact the route of delivery. -line 542 spell out TME The first citation of ‘tumor microenvironment (TME)’ occurs in line 129. Describe the G Advaccine.
We apologize, but we are not aware of a G Advaccine. Ad52 (the only member of the G subgroup), to the best of our knowledge, has not been described as a vaccine vector. If the Reviewer is referring to GVAX, while interesting, this approach (ex vivo cell modification) does not lend itself to the types of vector improvements and delivery routes that we aim to highlight.

Reviewer 3 Report

The authors provide a review on replication-deficient adenoviral vectors for cancer treatment, with special focus on immunotherapy and strategies to improve tumor targeting. These are relevant topics, which could be applied in part to replication-competent (oncolytic) adenoviruses. Since the review has a wide scope, good organization of data is essential. I recommend dividing section 2 (current applications) into some sub-sections focused on specific strategies (expression of immunostimulatory transgenes, tumor antigens, suicide genes, etc.).

Specific comments.

  1. The number of clinical trials using non-replicating adenoviral vectors for cancer therapy is very high. It is not clear which criteria have been used to incorporate/exclude clinical trials from table 1 and from the text. Just as an example, pioneering trials of adenoviral vectors encoding TK or IL-12 are missing. Among recent trials using the Ad-RTS-HIL-12 vector, only one of them is included in table 1, and the cited status (“discontinuation due to high toxicity”) does not match with the public releases of this trial.
  2. Also in table 1, it would be good to include the outcome of completed trials, and references, if available.
  3. The mechanism of adenovirus entry into cells is repeated in different sections of the review. Please eliminate redundancy.
  4. The concept of genetic modifications of adenoviral vectors to incorporate ligands in capsid proteins for further chemical coating should be discussed (combined genetic and chemical modifications).

Specific comments:

  1. In the simple summary, the following statement: “One of the fundamental approaches to cancer gene therapy is the use of non- replicating adenoviral vectors to deliver genes with anti-tumor activity” should be reconsidered. This might be true from the conceptual standpoint, but the clinical relevance of this approach is currently very limited.
  2. Recent references to anti Covid-19 vaccines using adenoviral vectors are missing.
  3. Line 57. The molecular biology of non-replicating and replicating adenoviruses is shared only on the initial steps of infection. Drastic differences are present afterwards (effects of E1 genes, genome replication, cytopathic effect, etc,).
  4. Line 343. Better explanation of the interaction of FX/Complement/Adenovirus is recommended.
  5. Line 379. Relative to reference #108, the accumulation in spleen refers to CD8 T cells, not vectors.
  6. The authors should acknowledge the discrepancies observed between CAR expression and infectivity of cells. In multiple cell lines and tissues, there is no such correlation.
  7. Line 675. The challenge of using adenoviral vectors for personalized immunotherapies should be mentioned. Development of a tailored vector for each patient is not clinically feasible. Alternatives should be discussed.

Minor points:

  1. Line 56. Revise “apply to applicable”.
  2. Line 66. Revise sentence.
  3. Line 178. Eliminate period (.)
  4. Line 385. Revise sentence.
  5. Line 426. Revise “[113] and coworkers”. The name of the first author should be mentioned.
  6. The same for line 668. “[178] demonstrated”.

Author Response

Tessarollo et al
Point-by-point response to Reviewers
#Reviewer 3: Comments and Suggestions for Authors The authors provide a review on replication-deficient adenoviral vectors for cancer treatment, with special focus on immunotherapy and strategies to improve tumor targeting. These are relevant topics, which could be applied in part to replication-competent (oncolytic) adenoviruses. Since the review has a wide scope, good organization of data is essential. I recommend dividing section 2 (current applications) into some sub-sections focused on specific strategies (expression of immunostimulatory transgenes, tumor antigens, suicide genes, etc.). We feel that dividing section 2 into sub-topics is complicated by the many approaches that use combinations of approaches. We also do not extensively explore each approach since our intention is to show a variety of possible modalities. Thus, sub-divisions may end up with a single example of the approach. Specific comments. The number of clinical trials using non-replicating adenoviral vectors for cancer therapy is very high. It is not clear which criteria have been used to incorporate/exclude clinical trials from table 1 and from the text. Just as an example, pioneering trials of adenoviral vectors encoding TK or IL-12 are missing. Among recent trials using the Ad-RTS-HIL-12 vector, only one of them is included in table 1, and the cited status (“discontinuation due to high toxicity”) does not match with the public releases of this trial. Thank you for the comments. We chose some to exemplify the different transgenes and approaches that could be used in several types of cancer, but did not intend to make a comprehensive list. We have added more examples (Ad-RTS-hIL-12) and, where possible, updated the status of the trials. Also in table 1, it would be good to include the outcome of completed trials, and references, if available. We added the available references. The mechanism of adenovirus entry into cells is repeated in different sections of the review. Please eliminate redundancy. Thank you for this observation. We consolidated the mechanism of adenovirus entry in section “3.1. Tissue Tropism” and only summarized this topic in lines 85-90. The concept of genetic modifications of adenoviral vectors to incorporate ligands in capsid proteins for further chemical coating should be discussed (combined genetic and chemical modifications).
Section 4.1 has been updated. We agree that this important topic should be addressed. Additional information was added (paragraph starting on line 714) showing describing genetic modification of the vector in order to achieve targeting or support chemical coating.
Specific comments: In the simple summary, the following statement: “One of the fundamental approaches to cancer gene therapy is the use of non- replicating adenoviral vectors to deliver genes with anti-tumor activity” should be reconsidered. This might be true from the conceptual standpoint, but the clinical relevance of this approach is currently very limited. We agree. The phrase now reads: “The treatment of cancer has progressed greatly with the advent of immunotherapy and gene therapy, including the use of non-replicating adenoviral vectors to deliver genes with anti-tumor activity for cancer gene therapy.” Recent references to anti Covid-19 vaccines using adenoviral vectors are missing. We have added this information. “These vaccines include those based on recombinant adenovirus serotypes as human adenoviral vector 5 [3–7], chimpanzee adenoviral vector ChAdOx1 [8,9] and combined human serotypes vectors 5 and 26 [10,11].” Lines 50 to 52. Line 57. The molecular biology of non-replicating and replicating adenoviruses is shared only on the initial steps of infection. Drastic differences are present afterwards (effects of E1 genes, genome replication, cytopathic effect, etc,). The paragraph starting at line 56 has been modified to avoid comparison between replicating and non-replicating vectors. The differences pointed out by the Reviewer are now highlighted in lines 115 to 118. Line 343. Better explanation of the interaction of FX/Complement/Adenovirus is recommended. We added the missing information in lines 399-401. Line 379. Relative to reference #108, the accumulation in spleen refers to CD8 T cells, not vectors. We apologize for this mistake. The phrase has been corrected. The authors should acknowledge the discrepancies observed between CAR expression and infectivity of cells. In multiple cell lines and tissues, there is no such correlation. We agree and have added the information about serotypes and the importance of their receptors in section “3.1 Tissue Tropism”, lines 342 to 350. We also added more information in the section “4.1. Modifications in viral entry: Attachment receptors and virus internalization” (lines 476-482). Line 675. The challenge of using adenoviral vectors for personalized immunotherapies should be mentioned. Development of a tailored vector for each patient is not clinically feasible. Alternatives should be discussed. We completely agree and have added some information about this important topic. Lines 752 to 759. Minor points:
Line 56. Revise “apply to applicable”. This phrase has been deleted. Line 66. Revise sentence. This phrase has been deleted. Line 178. Eliminate period (.) Done. Line 385. Revise sentence. This phrase has been modified. Line 426. Revise “[113] and coworkers”. The name of the first author should be mentioned. Done. The same for line 668. “[178] demonstrated”. Done.
